# Effect of Different Direct Compaction Grades of Mannitol on the Storage Stability of Tablet Properties Investigated Using a Kohonen Self-Organizing Map and Elastic Net Regression Model

**DOI:** 10.3390/pharmaceutics12090886

**Published:** 2020-09-18

**Authors:** Atsushi Kosugi, Kok Hoong Leong, Eri Urata, Yoshihiro Hayashi, Shungo Kumada, Kotaro Okada, Yoshinori Onuki

**Affiliations:** 1Formulation Development Department, Development & Planning Division, Nichi-Iko Pharmaceutical Co., Ltd., 205-1, Shimoumezawa Namerikawa-shi, Toyama 936-0857, Japan; a-kosugi@nichiiko.co.jp (A.K.); yoshihiro-hayashi@nichiiko.co.jp (Y.H.); shungo.kumada@nichiiko.co.jp (S.K.); 2Department of Pharmaceutical Chemistry, Faculty of Pharmacy, University of Malaya, Kuala Lumpur 50603, Malaysia; leongkh@um.edu.my; 3Laboratory of Pharmaceutical Technology, Graduate School of Medicine and Pharmaceutical Science for Research, University of Toyama; 2630 Sugitani, Toyama-shi, Toyama 930-0194, Japan; s1660203@ems.u-toyama.ac.jp (E.U.); kokada@pha.u-toyama.ac.jp (K.O.)

**Keywords:** excipient, storage stability, tablet, mannitol, Kohonen self-organizing map, Elastic net regression

## Abstract

This study tested 15 direct compaction grades to identify the contribution of different grades of mannitol to the storage stability of the resulting tablets. After preparing the model tablets with different values of hardness, they were stored at 25 °C, 75% relative humidity for 1 week. Then, measurement of the tablet properties was conducted on both pre- and post-storage tablets. The tablet properties were tensile strength (TS), friability, and disintegration time (DT). The experimental data were analyzed using a Kohonen self-organizing map (SOM). The SOM analysis successfully classified the test grades into three distinct clusters having different changes in the behavior of the tablet properties accompanying storage. Cluster 1 showed an obvious rise in DT induced by storage, while cluster 3 showed a substantial change in mechanical strength of the tablet including a reduction in the TS and a rise in friability. Furthermore, the data were analyzed using an Elastic net regression technique to investigate the general relationships between the powder properties of mannitol and the change behavior of the tablet properties. Consequently, we succeeded in identifying the crucial powder properties for the storage stability of the resulting tablets. This study provides advanced technical knowledge to characterize the effect of different direct compaction grades of mannitol on the storage stability of tablet properties.

## 1. Introduction

In manufacturing tablet products, a wide range of excipients are incorporated into the dosage form as well as the active pharmaceutical ingredients (APIs). The role of excipients is diverse (e.g., fillers, disintegrants, binders, and lubricants) [1]. Fillers are excipients to increase dosage form, volume, or weight, and then they are added to tablet formulations for the manufacture of properly sized products for easy handling. They also play an important role in improving tablet processability. Most fillers exhibit good compactability and increase the mechanical strength of tablets. Lactose is the most common substance used as a filler [1,2]. Recently, in the development of tablet products, the use of mannitol as a filler has increased. Although mannitol is more costly than lactose, it has numerous attractive properties as a filler [2,3]. For one thing, mannitol possesses high water solubility. In addition, mannitol provides patients with a pleasant cool feeling when it dissolves in the mouth. These properties are advantageous for a tablet filler, especially orally disintegrating tablets (ODTs). Furthermore, mannitol is more chemically inert than lactose, because mannitol is not accompanied by the Maillard reaction, which can occur between lactose and amine functional groups, in APIs, which leads to their degradation [4,5]. Thus, it can be used as an alternative filler to lactose.

To date, numerous commercial grades of mannitol have been produced by different manufacturers to deal with various demands in the manufacturing of tablets. As far as direct compression (DC) grade is concerned, there are various commercial grades. Each grade possesses different powder properties [4,6,7], which substantially affects the production and quality of the tablet products in a complex manner. Therefore, a comprehensive understanding of the effects of the different commercial grades on the properties of the resulting tablet has been very important for designing tablet formulations. Several researchers have investigated different mannitol grades in terms of the characterization of powder properties and the effect on tablet properties [4,6,8]. For example, Paul et al. examined 11 different grades of mannitol in terms of relationships between the powder properties (e.g., particle size, surface area, plasticity) and tableting performance (e.g., tensile strength of the resulting tablets) [4,6]. Similarly, Wagner et al. investigated the compactability of five different spray-dried mannitol grades [8]. Recently, we also investigated the powder properties of 15 different commercial DC grades of mannitol [9]. In the course of our study, a wide range of powder and tablet properties concerning the different grades were examined: the variables to be examined were the powder properties representing flowability, size, shape, and manufacturing properties, and tablet properties including disintegration time (DT), friability, and tensile strength (TS). Data analysis using a Kohonen self-organizing map (SOM) let us successfully characterize the test mannitol grades in terms of powder properties and thus identify the crucial powder properties for the resulting tablet properties.

In all likelihood, these differences in powder properties substantially affect not only the tablets’ initial properties but also their storage stability. However, no systematic comparative research of the different DC grades to investigate the change behavior of tablet properties provoked by storage has been performed to date. Against this background, this study focused on the effect of different mannitol grades on the storage stability of their tablet properties to gain further advanced technical knowledge about DC mannitol grades. This study tested the same 15 commercial mannitol grades as those in our previous study. The model tablets consisting of the different grades were stored under humid conditions. Then, the tablet properties (i.e., TS, friability, and DT) were measured pre- and post-storage. After that, the change in behaviors of the tablet properties induced by storage was characterized using the Kohonen SOM. In addition to the SOM analysis, general relationships of powder properties of the mannitol with the change in behaviors were investigated using the Elastic net (ENET) method. Using the ENET regression model, we identified the crucial powder properties for the storage stability of the tablet properties. The outcome of this study offers a comprehensive understanding of the effect of different DC grades of mannitol on the storage stability of tablet properties.

## 2. Materials and Methods

### 2.1. Materials

Fifteen different commercial DC grades of mannitol (Table 1) were used as purchased from the following suppliers: Mannit Q (MQ) from Mitsubishi Life Science Institute (Tokyo, Japan); Pearlitol 100SD (100SD), Pearlitol 200SD (200SD), Pearlitol 300DC (300DC), Pearlitol 400DC (400DC), and Pearlitol 500DC (500DC) from Roquette Japan (Tokyo, Japan); Parteck M100 (M100) and Parteck M200 (M200) from Merck Millipore (Billerica, MA, USA); Granutol F (GF), Granutol S (GS), and Granutol R (GR) from Freund (Tokyo, Japan); Mannogem EZ Spray Dried (EZ), Mannogem XL (XL), Mannogem 2080 Granular (2080), and Mannogem AG Granular (AG) from SPI Pharma (Wilmington, DE, USA). Microcrystalline cellulose (MCC) (Ceolus PH-302) was obtained from Asahi Kasei Chemicals (Tokyo, Japan). Low-substituted hydroxypropyl celluloses (L-HPC) (NBD-021) were provided by Shin-Etsu Chemical (Tokyo, Japan). Magnesium stearate (MgSt) was purchased from Fujifilm Wako Pure Chemicals (Osaka, Japan).

### 2.2. Preparation of Model Tablets

Model tablets containing different mannitol grades were prepared using the DC method. The formulation of the model tablets was as follows: 74% mannitol, 20% MCC, 5% L-HPC, and 1% MgSt. After sieving through a 20-mesh screen, designated amounts of ingredients were accurately weighed. These ingredient powders, except for the lubricant (MgSt), were placed in a polyethylene bag and then mixed for 3 min. After that, the MgSt was added to the mixture and then it was mixed again for 1 min. Three hundred grams of the powder blend were prepared for manufacturing each model tablet. The final blend (200 mg) was compressed into a flat round tablet, 8 mm in diameter, using a rotary tableting machine (VELA5; Kikusui, Kyoto, Japan). The model tablets for use in the storage experiment were prepared to have 55, 65, and 75 N of hardness by adjusting the compression force. In addition, model tablets were prepared with a compression force of 5 kN to evaluate the compression formability of the mannitol grades. The freshly prepared model tablets (pre-storage tablets) and those stored at 25 °C, 75% RH for 1 week (post-storage tablets) were used for the measurement of tablet properties described below. This storage condition was adopted based on our previous study [10].

### 2.3. Tensile Strength

The hardness of the tablets was determined using a tablet hardness tester (Ogawa Seiki, Tokyo, Japan). The TS was calculated as
(1)$$\mathrm{Tensile}\;\mathrm{strength}\;\left(\mathrm{MPa}\right)=\frac{2F}{\pi dt}$$
where *F* is the hardness (the maximum diametric crushing force), and *d* and *t* are the diameter and thickness of the tablet, respectively.

### 2.4. Friability

The friability of tablets was determined using a friability tester (TFT-1200; Toyama Sangyo, Osaka, Japan). For each measurement, ca 4.0 g of the tablets was weighed and then placed in a friability tester. Afterward, the drum of the friability tester was rotated at 25 rpm for 10 min. The tablet powder was collected and then sieved through a 16-mesh screen (aperture size: 1 mm). The friability was calculated as the weight ratio of the tablets before and after rotation.

### 2.5. Disintegration Time

For the measurement of the DT of the model tablets, this study employed an ODT testing apparatus (ODT-101; Toyama Sangyo). Purified water at 37 ± 0.5 °C was employed as the test medium. A detailed explanation of this apparatus has been given elsewhere [9]. In short, a test tablet was placed on a stainless-steel plate having many small holes in a regular pattern and the bottom surface of the plate was faced with the test medium filled in the vessel. To start the measurement, the tablet was pressed with a column-shaped metallic weight (20 g) whose bottom surface was covered with a soft sponge. The metallic weight was connected to a rotation shaft and then it was rotated at a speed of 100 rpm during the measurement. Depending on the rotational stress, degradation of the tablet gradually developed. The disintegrated portions were removed from the tablet by passing through the small holes of the stainless-steel plate. The time required for the tablet to disappear completely on the plate was defined as DT. The measurement of DT was precisely carried out by detecting the change in electrical resistance of the bottom surface of the metallic weight and the stainless-steel plate.

### 2.6. Data Analysis

Viscovery SOMine^®^ (Version 7; Eudaptics Software GmbH, Vienna, Austria) was employed for the Kohonen SOM clustering. The Kohonen SOM is a feedforward-type neural network model [11]. This analysis allows multidimensional data to be turned into a two-dimensional map. This is considered to be a powerful tool for characterizing multidimensional data, especially clustering data. The typical structure of a SOM is composed of one input layer and one output layer. Nodes containing parametric reference vectors are arranged in a regular pattern in the output layer. The SOM is constructed via an unsupervised competitive learning approach. As a result of competitive learning, the array of the output nodes ultimately reflects the patterns of learning data sets. The adjacent nodes possess closer reference vectors to each other; thus, the distance between the nodes can be regarded as the degree of similarity between them. To conduct the SOM clustering, the number of nodes was set at 300 and the following nine combinations of tablet properties were used as the training variables: % retentions in TS, friability, and DT of the model tablets using 55, 65, and 75 N of hardness. The competitive learning process was iterated to generate the SOM. The number of iterations of competitive learning is automatically optimized by the software. For this SOM clustering, the learning of the data set was iterated 17 times.

ENET regression was carried out using JMP^®^ Pro 14 statistical software (SAS Institute, Cary, NC, USA). ENET is the latest sparse modeling method, which is similar to the Lasso and ridge regressions [12,13]. Sparse modeling methods feature regularized or penalized regression techniques. Better regression models are constructed by shrinking the model coefficients toward zero. The validity of the estimated regression models was tested using Bayesian information criterion. These models have a great capacity to identify the crucial factors for characteristics from data sets consisting of a large number of variables. Additionally, the analysis is robust against confounding bias compared with conventional regression methods. Recently, sparse modeling methods including ENET have attracted great attention as being powerful tools for the characterization of multidimensional data. Some technical reports have applied ENET in pharmaceutical research [14,15].

## 3. Results and Discussion

### 3.1. SOM Analysis to Characterize the Change Behaviors of Tablet Properties of the Model Tablets Induced by Storage for One Week

The DC grades of mannitol tested in this study are summarized in Table 1. The spray-dried mannitol grades correspond to MQ, 100SD, 200SD, EZ, XL, M100, and M200, while granulated grades were 300DC, 400DC, 500DC, 2080, AG, GF, GS, and GR [4,6,7]. In addition, they are different in terms of crystalline forms: namely, 100SD, 200SD, EZ, and XL were a mixture of α and β crystalline forms, while the other grades consisted of the β form only. The microscopic images of these powders were shown in Appendix A (see Appendix A).

The model tablets consisting of each grade were prepared to adjust their tablet hardness to 55, 65, and 75 N. Afterward, the tablet properties were measured pre- and post-1-week storage at 25 °C, 75% RH. All components of the model tablets are stable against humid storage conditions. We thought the values of the tested tablet properties mainly fluctuated according to the moisture content of the tablets. In other words, these values should be constant as long as the moisture content reaches a level of saturation after storage under humid conditions [16,17,18]. For example, Yamazaki et al. [18] studied the storage stability of original brand name and generic famotidine tablets after storage under humid conditions (27 °C, 55% RH) for 8 weeks. The hardness and DT dropped immediately after storage, but these variables appeared to be constant afterwards until the end of the experimental period. Sakamoto et al. [16] monitored time-dependent change in tablet properties of magnesium oxide ODTs following an accelerated stress test (40 °C, 75% RH). They also reported that TS dropped sharply shortly after beginning the accelerated storage test, and afterwards the values remained constant for 4 weeks [16]. Regarding our model tablets, we previously confirmed that the moisture content reached a level of saturation within the first week of storage under humid conditions [10]. Thus, the change in the tablet properties after 1 week of storage was used to evaluate its storage stability.

As we predicted, overall TS values were reduced by storage (Figure 1A). The degree of decrease in the TS substantially changed according to the mannitol grades and hardness. In particular, 300DC-, 400DC-, 500DC-, 2080-, and AG-containing tablets showed obvious reductions. Figure 1B represents the deltas of variables between pre- and post-storage tablets as a function of tablet hardness. The mean value, expressed as × in the box plots, decreased significantly with higher hardness, while the variances between the first quartile and third quartile were hardly changed even though the hardness was changed. The changes in friability behavior showed a significant negative correlation with those of TS: a more substantial reduction in TS after 1 week of storage, and a more substantial rise in friability. The 400DC-containing tablets showed the most significant changes in friability and TS after storage. The delta values of friability as a function of hardness were different from those of TS: The mean and variance obtained from the tablets with different hardness were very close to each other. As for DT, on the whole, the values were increased with higher tablet hardness. Most of the model tablets resulted in an increase in DT on storage, while some test tablets (e.g., 400DC-containing tablets with 65 and 75 N) showed a decrease in DT. Thus, the change in behavior of DT seemed more complicated than those of TS and friability. In addition, tablet hardness appeared to affect the change behavior of DT. That is because, although the mean values of the delta of DT (Figure 1B) were almost the same, their variances gradually increased with higher tablet hardness.

Subsequently, we characterize the changes in the tablet properties of each model tablet after storage. We calculated the % retention of the tablet properties from the pre-storage level as an indicator to evaluate the storage stability of each tablet. As shown in Figure 1, tablet properties, especially TS and DT, were changed proportionally according to the tablet hardness. Since the % retention is a parameter that removes the influence of absolute values of tablet properties from the raw data, we assumed that the all data can be evaluated regardless of the tablet hardness by using this parameter. The calculation of the % retention of the tablet properties from the pre-storage level was performed as follows:(2)$$\%\;retension\;of\;tablet\;properties\;from\;the\;prestorage\;level=\;\frac{x_{1week}}{x_{pre}}\times 100$$
where *x*_1*week*_ is the value for the variables of the post-storage tablets, while *x*_*pre*_ is that of the pre-storage tablets. The *x*_*pre*_ corresponds to the mean value of each grade of mannitol tablet for the variable measurement, which was repeated three times. A higher % retention means that the value of the tablet property is increased more significantly due to storage for 1 week. As shown in Figure 2A, a similar change in behavior as a function of mannitol grades was observed from tablets with different levels of hardness. In accordance with this, the statistics of % retention as a function of hardness were almost the same in terms of mean and variance (Figure 2B). We conclude that, as far as this study is concerned, % retention from the pre-storage level enables us to extract the essence of the effect of mannitol grades on the storage stability of tablet properties.

The calculated % retention values were analyzed using Kohonen SOM to characterize the change behaviors in tablet properties following storage. Kohonen SOM enables to express multidimensional data as a two-dimensional map, which is a powerful tool for clustering data. SOM analyses have been applied in the pharmaceutical and medical fields [19,20,21]. The present study tried to classify the test mannitol grades into several clusters in terms of change behavior of tablet properties accompanying the storage. The following nine variables were employed as training variables of the SOM clustering: % retentions from the pre-storage levels in TS, friability, and DT of the model tablets having 55, 65, and 75 N of hardness. The SOM algorithm is based on competitive learning [11]. The SOM shown in Figure 3A was obtained as a result of the competitive learning. The constituent unit of SOM is called a “node.” Numerous nodes are arranged in a regular pattern in the SOM. Each node possesses parameter information, which is the same data size as the input data. In SOM, the adjacent nodes are supposed to have similar properties; thus, the distance between the nodes expresses the degree of similarity. As a result of SOM clustering, the model tablets containing different mannitol grades were classified into three distinct clusters (Figure 3A). Each experimental datum used for the SOM clustering was labeled on the nodes of the resulting SOM: 100SD-, 200SD-, GR-, M200-, MQ-, and XL-containing tablets were assigned to cluster 1; EZ-, GF-, GS-, and M100-containing tablets were assigned to cluster 2; and 300DC-, 400DC-, 500DC-, 2080-, and AG-containing tablets were assigned to cluster 3. Figure 3B shows a representative result of the SOM feature maps. They correspond to the % retention of tablet properties measured from the model tablets formed at 65 N. As shown in this figure, the SOM feature maps display the regions distributing higher and lower values of the variables in red and blue. From the SOM feature maps, one can easily see the latent relationships among the variables. This study also summarized the % retention of tablet properties for each grade of the clusters to characterize their different storage stabilities (Figure 4). Regarding TS, a substantially low % retention was observed for cluster 3, while the values of clusters 1 and 2 showed less decrease. This indicates that TS of cluster 3 was substantially decreased due to storage, while those of clusters 1 and 2 showed less change. The change behavior of friability appeared to be strongly associated with that of TS. That is because a substantially high % retention of friability was observed for cluster 3, while those of clusters 1 and 2 were hardly changed after storage. It is worth noting that % retention in TS of the entire data had a highly negative correlation with that of friability (*r* = −0.906) (see Appendix A, Appendix A). As for DT, the change behavior induced by storage was independent of TS and friability: The correlation coefficients (*r*) of % retention of DT compared with those in TS and friability were very low, 0.501 and −0.418, respectively (Appendix A). Unlike TS and friability, cluster 1 showed an obvious rise in DT after storage. By contrast, the data for DT in clusters 2 and 3 were mainly distributed around 100% (close to the prestorage level); thus, these clusters showed less change in terms of DT. From these findings, we succeeded in characterizing the change behaviors of tablet properties accompanying storage.

### 3.2. Relationships of Powder Properties of Mannitol Grades with the Storage Stability of the Tablet Properties Using an ENET Regression Model

We further characterized the powder and tablet properties of the clusters. We have previously measured various powder properties of the test mannitol grades [9]. This study used the following six powder properties: bulk density (*V*_bulk_), compressibility index (CI), angle of repose (AR), median diameter of particle size (D50), specific surface area (SSA), and ejection stress (ES). ES is a manufacturing property measured using a powder compaction analysis system (GTP-2; Gamlen Instruments, London, UK) [22,23,24]. As for friability and DT, the values of the model tablets with a hardness of 65N were employed as representative data. Additionally, the compression force was recorded in preparing the model tablets; thus, the compression force values were also used for this experiment. As for TS, we prepared the model tablets under the same compression force of 5 kN and then used their TS values for this experiment. To compare variables between the clusters, the standardized differences were calculated as follows (Figure 5A):(3)$$Standardized\;difference=\;\frac{\left({\bar{X}}_{cluster}-\bar{X}\right)}{SD}$$
where ${\bar{X}}_{cluster}$ is the mean value of the variables of mannitol grade assigned in each cluster. $\bar{X}$ and *SD* are the mean and standard deviation of the entire data set, respectively. We further compared the data for each cluster with the entire data using the Student *t* test, and then determined significant differences in the variables for each cluster. From this analysis, the features of the powder and tablet properties of each cluster are fully understood. Cluster 1 contains 300DC, 400DC, 500DC, 2080, and AG. They are granulated mannitol grades (Table 1). The unique properties were observed as follows: higher TS, shorter DT, lower compression force, and lower *V*_bulk_ and D50. Cluster 2, which was the most stable against storage, showed higher TS, lower compression force, higher CI, lower D50, and higher ES. The mode of action of the variables of cluster 2 was very similar to that of cluster 1, but the variables were distinguishable in terms of the significant differences in DT, *V*_bulk_, CI, and ES. Cluster 3 showed a distinct mode of action compared with the other clusters: lower TS, higher friability, shorter DT, higher compression force, higher *V*_bulk_, D50, and lower CI, SSA, and ES. In addition, the SOM feature maps of the powder and tablet properties are presented in Figure 5B. Since the input data for the SOM clustering contained not only training variables (% retention values), but also powder and tablet properties, their SOM feature maps could be created from the SOM constructed before. Based on the SOM feature maps, we can visually understand the relationships between powder and tablet properties. For example, the SOM feature maps clearly show that mannitol grads of cluster 3 possess higher D50 and lower CI and SSA. Furthermore, cluster 3 appeared to have less compactability because they require higher compression force to prepare the tablets than the other clusters. These relationships are consistent with relevant comparative studies [6,8]. Paul et al. [6] reported that granulated mannitol (i.e., 300DC, 400DC and 500DC) exhibited poorer compactability than those of spray-dried mannitol (i.e., M100, M200, 100SD and 200SD). The authors mentioned that the granulated mannitol has powder properties of larger particle size, smaller SA and lower plasticity, and then these powder properties lead to a smaller bonding area of particles during compression, resulting in poor compactability. Wagner et al. [8] also tested compactability of five different spray-dried grades of mannitol, and then reported that mannitol grades having higher SA are prone to produce tablets having higher TS.

We further investigated the general effect of the powder properties of mannitol on the storage stability of the resulting tablets. The analysis was performed using ENET regression modeling. According to the analysis, we tried to identify the crucial powder properties for the % retention from the pre-storage level in the tablet properties. It is worth noting that ENET is robust against the confounding effect of factors. Since some powder properties of mannitol were highly correlated, we considered it to be suitable for the analysis. Before conducting the ENET analysis, the powder properties were standardized so that a relative comparison of their effects could be performed.

Figure 6 shows scatterplots of experimental versus predicted values for tablet properties calculated using the constructed regression models. The obtained ENET regression models for tablet properties are listed in Table 2, Table 3 and Table 4. The predicted values in the scatterplots (Figure 6) were estimated from the obtained ENET regression models. The correlation coefficients (*r*) of the regression models concerning TS and friability were extremely high: The coefficients for % changes in TS and friability were 0.923 and 0.911. The crucial powder properties were identified by the regression models (Table 2, Table 3 and Table 4). We note the same factors (i.e., *V*_bulk_, D50, and SSA) were proved to have a significant effect on % retention of TS and friability (Table 2 and Table 3). Based on the result that a lower % retention of TS and a higher % retention of friability represent a more significant change according to the storage (Figure 4), mannitol, having larger *V*_bulk_ and D50 and smaller SSA, is prone to produce tablets whose TS and friability are susceptible to storage. We further note that a higher χ^2^ value of the explanatory variable represents a more significant effect. Thus, D50 has by far the most significant impact on changes in TS and friability. As mentioned in Figure 5B, mannitol grades with larger D50 and smaller SSA required higher compression force to prepare tablets: thus, such a low compactability might contribute to significant changes in the mechanical strength of the tablets after storage. As for the change behavior of DT, *V*_bulk_, CI, and D50 were found to be significant factors (Table 4). In particular, the effect of *V*_bulk_ was the most significant. Namely, smaller *V*_bulk_, CI, and D50 of the mannitol are prone to lead to a significant prolongation of DT due to storage. From these findings, we successfully characterized the effect of different mannitol grades on the change behaviors of tablet properties induced by storage.

## 4. Conclusions

The present study focused on the effect of different commercial DC grades of mannitol on the storage stability of tablet properties. SOM clustering successfully classified the 15 different mannitol grades into three distinct clusters according to the change behaviors of the tablet properties. Cluster 1 contained 100SD, 200SD, GR, M200, MQ, and XL. Their tablets showed a significant increase in DT after 1 week of storage. Cluster 2, containing EZ, GF, GS, and M100, was the most stable for all variables among the test mannitol grades. Cluster 3 contained 300DC, 400DC, 500DC, 2080, and AG. These tablets showed a significant change in mechanical strength including a significant reduction in TS and a significant rise in friability on storage. This study proceeded with an investigation of the general relationships of the powder properties to the change behavior of the tablet properties using the ENET technique. In consequence, we identified the crucial powder properties for the storage stability of tablet properties. In conclusion, this study provided improved knowledge of the effect of different mannitol grades on the stability of the resultant tablet properties. This knowledge is valuable information for designing tablet formulations.

## Figures and Tables

**Figure 1 pharmaceutics-12-00886-f001:**
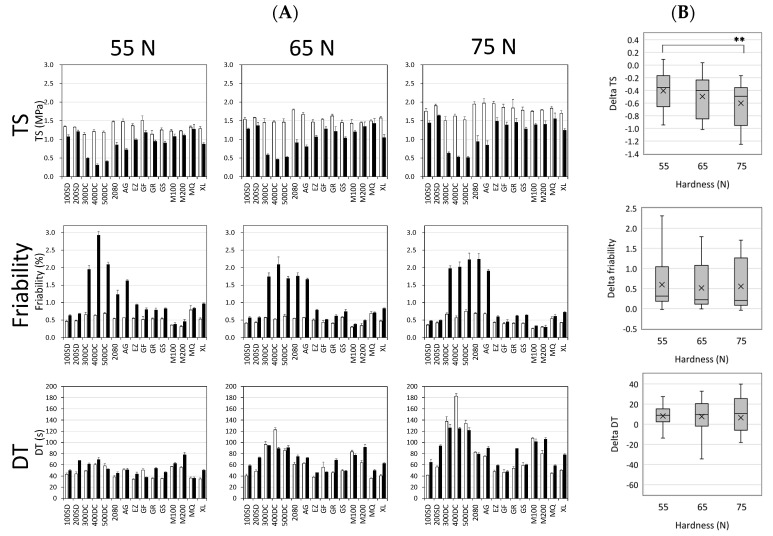
Tablet properties of pre- and post-storage tablets consisting of different mannitol grades. Model tablets having 55, 65, and 75 N of hardness were tested. (**A**) White and black bars represent the variables of pre- and post-storage tablets, respectively. Each bar represents the mean ± SD (*n* = 3). (**B**) Box plots of the delta of variables between pre- and post-storage tablets. The lower and upper ends of the rectangular box represent the first and third quartiles of the variables, a horizontal line and “×” in the interior of the box express the median and mean, and the lower and upper ends of the whiskers indicate the minimum and maximum of the variables, respectively. ** *p* < 0.01.

**Figure 2 pharmaceutics-12-00886-f002:**
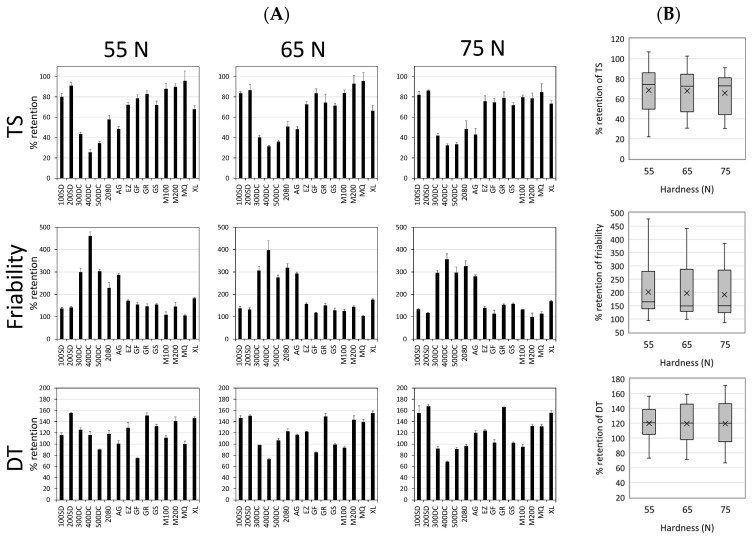
Change in tablet properties induced by storage for 1 week at 25 °C, 75% RH. (**A**) % retention of tablet properties measured pre- and post-1-week storage. The model tablets consisting of 74% individual mannitol grades were prepared to have 55, 65, and 75 N of hardness. Each bar represents the mean ± SD (*n* = 3). (**B**) Box plots of the % retention of the tablet properties as a function of hardness. The lower and upper ends of the rectangular box represent the first and third quartiles of the variables, a horizontal line and “×” in the interior of the box express the median and mean, and the lower and upper ends of the whiskers indicate the minimum and maximum of the variables, respectively.

**Figure 3 pharmaceutics-12-00886-f003:**
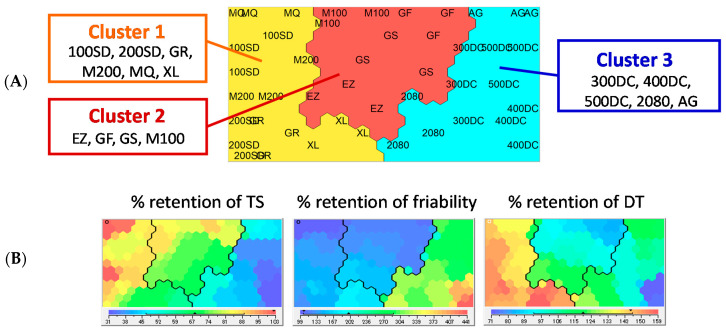
(**A**) SOM and (**B**) SOM feature maps to classify test grades of mannitol according to the storage stability of tablet properties. The SOM clustering was performed using % retention of TS, friability, and DT of tablets having 55, 65, and 75 N of hardness as training variables. Each experimental data set was labeled on the nodes of the map to express the data distribution (**A**). The properties of the tablets having 65 N of hardness are shown in the SOM feature maps as a representative variable (**B**).

**Figure 4 pharmaceutics-12-00886-f004:**
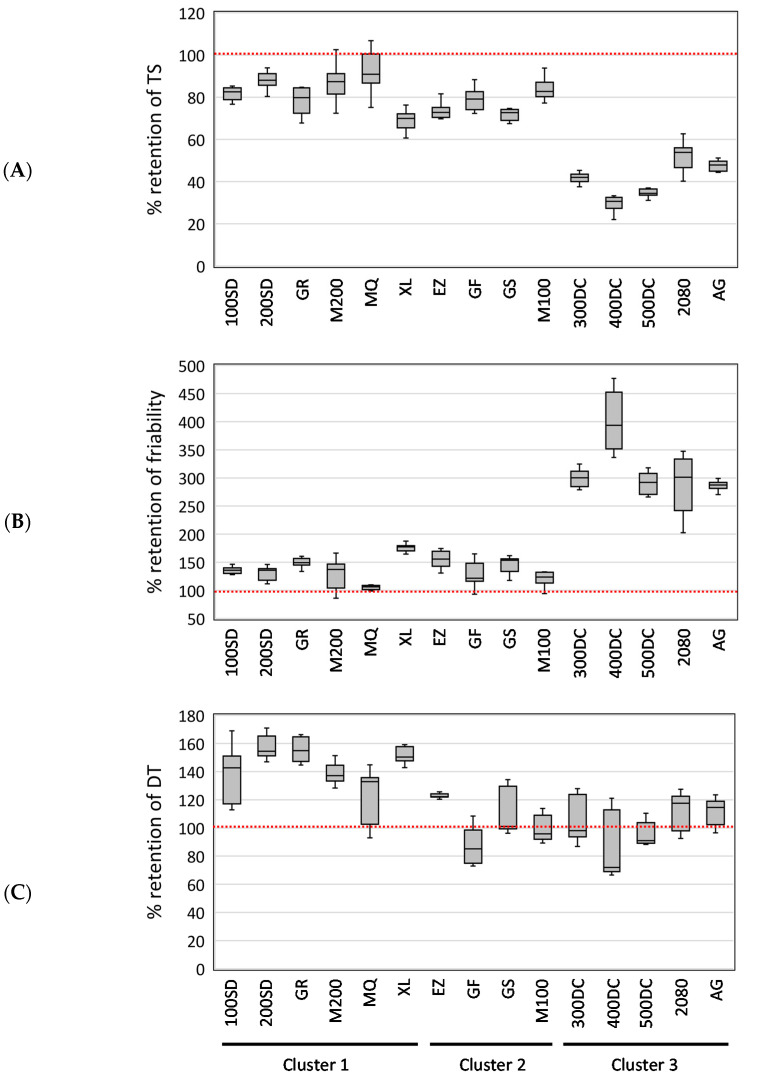
Box plots of the % retention from the pre-storage levels of TS (**A**), friability (**B**), and DT (**C**) as a function of mannitol grades. The lower and upper ends of the rectangular box represent the first and third quartiles of the variables, a horizontal line in the interior of the box expresses the median, and the lower and upper ends of the whiskers indicate the minimum and maximum of the variables, respectively.

**Figure 5 pharmaceutics-12-00886-f005:**
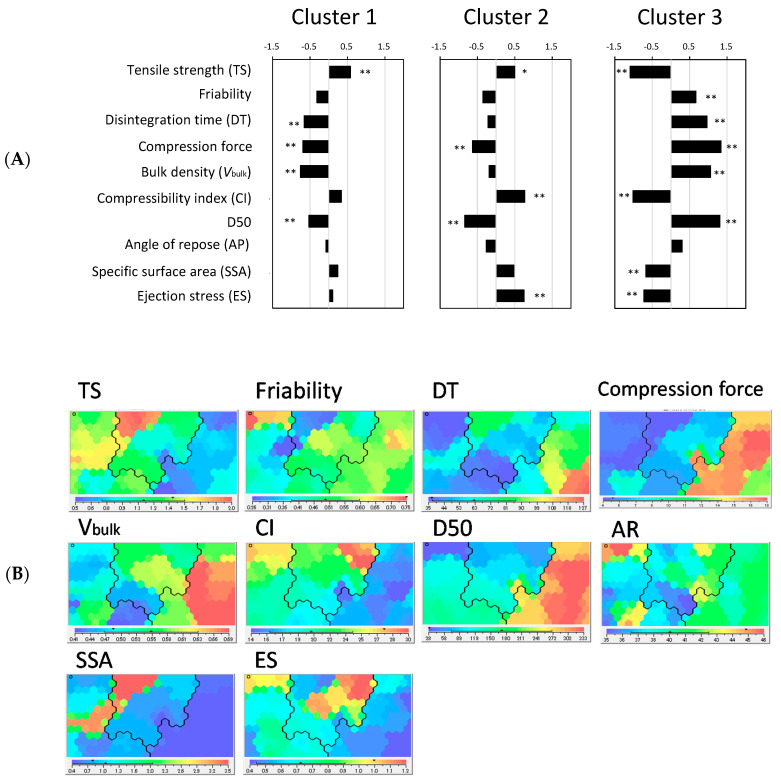
Powder and tablet characteristics of the SOM clusters. (**A**) Profiling analysis of SOM clusters. Each value represents the standardized differences in the means of variables between each cluster and the entire data set calculated according to Equation (3). The TS data were acquired from the model tablets prepared at a compression force of 5 kN, while those of friability and DT, compression force (kN) were from the model tablets with 65 N of hardness. The data of powder properties including *V*_bulk_, CI, AR, D50, SSA, DS, and ES are quoted from the previous study [9]. ** *p* < 0.01 and * *p* < 0.05 vs. the entire data. (**B**) SOM feature maps were created by arranging the data according to the constructed SOM shown in Figure 3.

**Figure 6 pharmaceutics-12-00886-f006:**
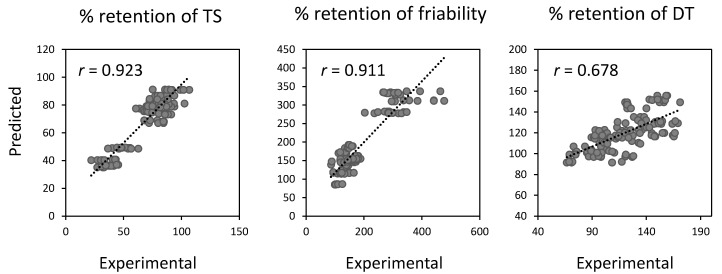
Scatterplots of experimental vs. predicted values. The predicted variables were obtained from the regression models estimated using the ENET regression technique.

**Table 1 pharmaceutics-12-00886-t001:** Direct compaction grades of mannitol tested in this study.

Mannitol Grades	Abbreviation	Manufacturing Method	Crystalline Form
Mannit Q	MQ	Spray-drying	β
Pearlitol 100SD	100SD	Spray-drying	α, β
Pearlitol 200SD	200SD	Spray-drying	α, β
Pearlitol 300DC	300DC	Granulation	β
Pearlitol 400DC	400DC	Granulation	β
Pearlitol 500DC	500DC	Granulation	β
Parteck M100	M100	Spray-drying	β
Parteck M200	M200	Spray-drying	β
Granutol F	GF	Granulation	β
Granutol S	GS	Granulation	β
Granutol R	GR	Granulation	β
Mannogem EZ Spray Dried	EZ	Spray-drying	α, β
Mannogem XL	XL	Spray-drying	α, β
Mannogem 2080	2080	Granulation	β
Mannogem AG	AG	Granulation	β

**Table 2 pharmaceutics-12-00886-t002:** Crucial powder properties affecting % retention of TS estimated using the ENET technique.

Factor	Estimate ^1^	Standard Error ^1^	χ^2^ Value	*p* Value
Bulk density	−55.4	11.3	23.9	<0.01
D50	−162.6	11.4	201.6	<0.01
Specific surface area	26.0	9.3	7.7	<0.01

^1^ Estimates of standardized regression coefficients and standard errors of the regression model.

**Table 3 pharmaceutics-12-00886-t003:** Crucial powder properties affecting % retention of friability estimated using the ENET technique.

Factor	Estimate ^1^	Standard Error ^1^	χ^2^ Value	*p* Value
Bulk density	178.9	42.9	17.4	<0.01
D50	792.7	41.7	360.9	<0.01
Specific surface area	−65.1	27.0	5.8	0.016

^1^ Estimates of standardized regression coefficients and standard errors of the regression model.

**Table 4 pharmaceutics-12-00886-t004:** Crucial powder properties affecting % retention of DT estimated using the ENET technique.

Factor	Estimate ^1^	Standard Error ^1^	χ^2^ Value	*p* Value
Bulk density	−185.6	26.1	50.4	<0.01
Compressibility index	−167.8	42.2	15.8	<0.01
D50	−99.9	49.2	4.1	0.042

^1^ Estimates of standardized regression coefficients and standard errors of the regression model.

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
