# Peer review of "Effect of Different Direct Compaction Grades of Mannitol on the Storage Stability of Tablet Properties Investigated Using a Kohonen Self-Organizing Map and Elastic Net Regression Model"

_pharmaceutics, 2020, doi:10.3390/pharmaceutics12090886_

Round 1

Reviewer 1 Report

The authors have extended their existing work on the tableting properties of direct compaction grades of mannitol from different suppliers  to assess differences in changes in  properties  during stability storage. The work is of interest to those interested in the utility of these material in developing formulations of orally disintegrating tablets and other solid dosage forms where mannitol would be a desired excipient.

Before the manuscript can be considered further for possible publication, there are a few areas where the authors need to make corrections or add in further information to more completely explain their findings, as noted below.

Page 1 line 40-41:  fillers are indeed added to make for properly sized tablets for easy handling, but they add other functionality also,  for example to improve processability  in terms of yielding adequately hard tablets to withstand handing during further processing (e.g. packaging).As changes in hardness/tensile strength are one aspect the authors have considered in their work, this mention of fillers also possibly providing for desired level of hardness/tensile strength ought to be introduced here. 

Page 2, line 49: authors might want to restate how big a concern Maillard reaction is.  I would  agree it is something to be properly considered but may not be  a "major" concern as its development can also be controlled by  moisture content and storage condition.

Page 2 line 70-71: authors should indicate as to why they chose these stress conditions, did they have  preliminary data to suggest they would be appropriate? They do not seem to be an overt degree of stress.

Page 3 line 99-102: authors must indicate more clearly  how they did mixing.  States it was done by shaking a polyethylene bag with the ingredients to be mixed  contained therein.  I find it hard to accept they did this on the 300kg scale indicated.

Page 3 line 103- 106: authors must indicate solid fraction produced by compressing the different mannitol blends.  This data (to be put into the results and discussion) may help further explain how differences in mannitol particle properties result in tablets with different initial and storage properties when compressed to specific hardness.

Page 3 line 111: confirm equation is correct. has ?? symbols in front of dt in denominator and the symbol t is given as t'

Page 11 line 317: notes that D50  has most impact on changes in tablet properties, which might align with my earlier comment that the solid fraction to which tablets are compressed might be important not the hardness to which they are compressed, as  solid fraction for a given applied force might correlate with particle size (e.g. D50).

Author Response

We thank the reviewers for the valuable comments on our manuscript. The manuscript has been revised to address the reviewers’ concerns. The sentences that were revised are highlighted in yellow. A list of responses to the reviewers’ comments follows.

Responses to Reviewer #1’s comments

Page 1 line 40-41:  fillers are indeed added to make for properly sized tablets for easy handling, but they add other functionality also, for example to improve processability in terms of yielding adequately hard tablets to withstand handing during further processing (e.g. packaging).As changes in hardness/tensile strength are one aspect the authors have considered in their work, this mention of fillers also possibly providing for desired level of hardness/tensile strength ought to be introduced here. 

RE: We appreciate the valuable comment. We revised the text as follows.

Page 1, Lines 42-43

They also play an important role in improving tablet processability. Most fillers exhibit good compactability and increase the mechanical strength of tablets.

Page 2, line 49: authors might want to restate how big a concern Maillard reaction is.  I would agree it is something to be properly considered but may not be a "major" concern as its development can also be controlled by moisture content and storage condition.

RE: This statement was also pointed out by the other reviewer. We took account of these comments and revised the text as follows.

Page 2, Line 49-51

Furthermore, mannitol is more chemically inert than lactose, because mannitol is not accompanied by the Maillard reaction that can occur between lactose and amine functional groups in APIs leads to their degradation [4, 5]. Thus, it can be used as an alternative filler to lactose.

Page 2 line 70-71: authors should indicate as to why they chose these stress conditions, did they have preliminary data to suggest they would be appropriate? They do not seem to be an overt degree of stress.

RE: The storage condition (25oC, 75%RH) was decided based on our previous study. This storage condition is commonly used for storage tests in the Japanese pharmaceutical industry. The condition is likely to be determined by considering humid climate of Japan. We revised the text as follows.

Page 3, Lines 114-115

This storage condition was adopted based on our previous study [10].   

Page 3 line 99-102: authors must indicate more clearly how they did mixing.  States it was done by shaking a polyethylene bag with the ingredients to be mixed contained therein.  I find it hard to accept they did this on the 300kg scale indicated.

RE: We apologize that there is mistake in the text of sample preparation. Correctly, three hundred “grams” of the powder blend were prepared for each model tablets. Thank you for the correction of my mistake.

Page 3 line 103- 106: authors must indicate solid fraction produced by compressing the different mannitol blends.  This data (to be put into the results and discussion) may help further explain how differences in mannitol particle properties result in tablets with different initial and storage properties when compressed to specific hardness.

RE: We have already presented SEM images of the mannitol grade particles in our previous article (J. Pharm. Sci. 2020, 109, 2585–2593). This time, the particles of test mannitol grades were observed again using a digital microscope, and then the microscopic images were shown as supplemental material (see, Fig.S1).

Page 3 line 111: confirm equation is correct. has ?? symbols in front of dt in denominator and the symbol t is given as t'

RE: Thank you for your pointing out my mistake. We deleted the comma from the equation (1).

Page 11 line 317: notes that D50  has most impact on changes in tablet properties, which might align with my earlier comment that the solid fraction to which tablets are compressed might be important not the hardness to which they are compressed, as  solid fraction for a given applied force might correlate with particle size (e.g. D50).

RE: We have recorded the compression force in preparing the model tablet. As the reviewer expected, D50 was associated with compression force. By considering this issue, we added the data of compression force to Fig. 5. Also, contribution of D50 and compression force to storage stability of TS and friability was discussed in the text as follows.

Page 12, Lines 373-375

As mentioned in Fig.5B, mannitol grades with larger D50 and smaller SSA required larger compression force to prepare tablets: thus, such a low compactability might contribute to significant changes in the mechanical strength of the tablets after storage.

Reviewer 2 Report

In the submitted manuscript authors have presented a study on the effect of different compaction grades mannitol on the storage stability of tablet properties. A Kohonen self-organizing map and elastic network regression model were used to analyze the data.

The concept is overall clearly presented; however, I have some concerns regarding the duration of storage for assessment of the stability. In my opinion, one week is a too short period of test for any general conclusion regarding the long-term tablets’ stability. On the other hand, if tablets are prone to succumb to significant changes in mechanical properties and/or disintegration time after just one week then one has to wonder what would happen after several months or years of storage.

The abstract ends with the statement that “this study could offer insight into the characteristics f mannitol”. This is too vague, authors should be more precise on what specific properties of mannitol, or properties of tablets that are made with different grades of mannitol, have been identified.

One general comment is that authors should omit over self-citations and provide more comparisons and/or correlations to studies of other research groups, especially in the context of properties of mannitol.

In the introduction section authors have highlighted the benefits of mannitol in the formulation of orally disintegrating tablets (ODTs) as well as its superiority over lactose. Some sentences are repetitive, especially concerning comparison to lactose, therefore this whole paragraph should be shortened. In addition, the whole first part of the introduction section is a repetition, although somewhat reformulated, of the authors’ previous publication.

In line 56, I suggest authors list different powder properties of directly compressible commercial grades of mannitol. Also, authors should highlight what are the critical quality attributes of ODTs in general, in order to avoid the reflection of their previous paper to a great extent.

Authors should include additional references, apart from their previous study, that are relevant for the presented findings. Have there been other reports on the stability testing where the samples have been conditioned for just one week? What about the selection of algorithms/models for the data analysis? From the current perspective, it seems that they were somehow randomly chosen. I suggest authors introduce more carefully the available algorithms and discuss the properties of the dataset size and data attributes that allow the usage of specific methods for data analysis. If other data analysis methods were used, even if the outcome was unsuccessful, it should be interpreted as well in order to help and provide guidance for their potential future applications for similar studies.

The introduction section should contain more relevant references regarding the issues on compaction properties of mannitol, stability studies, and data analysis methods, rather than already known properties of lactose and ODTs. In the current version, the last paragraph of the introduction section is just a recapitulation of what the authors have done and as such is not acceptable.

Authors should comment on the formulation composition – why were MCC and L-HPC used as additional excipients? What about the superdisintegrant(s) that are usually used for ODTs, and that are known that can potentially impact both mechanical properties and the disintegration time of tablets?

Is there any specific reason for manufacturing at such a large scale – three hundred kilograms per batch?

There are some technical issues with Eq. 1.

How were the parameters for the friability test selected?

Authors have stated, in line 137, that there was triplicate data acquisition for 15 different grades of mannitol. Triplicate would mean that the same conditions were used for each batch for three times. However, each batch was compressed at different compression pressures to produce three different tablets’ hardness values. I suggest authors to define more precisely what was varied and what was repeated.

The authors should state the number of tablets per batch that were used to test tensile strength, friability, and disintegration time. In line 186 it is stated that measurements were made in triplicate which is insufficient, especially in the case of disintegration time and friability.

The authors should provide more details on the parameters of the elastic regression network model that was used.

A comparison between the results of mannitol tablets characterization with the relevant references (other from authors’ own) should have been made.

What is the expected reason for the reduction in tablets’ tensile strength upon one week of storage? Are there any other reports on the same matter? Why did tablets made of 400DC have the highest change in tablets tensile strength and friability upon storage?

A proper discussion of the obtained results, in terms of the effect of storage on tablets properties, should contain proposed mechanisms and/or explanations of the obtained findings. In the current version of the manuscript, authors are just describing in more detail the results that are presented in Figure 1.

I disagree with the authors that the introduced measure in Eq.2 removes the influence of the tablet hardness on the obtained data. It just represents the % of retention of a tablet property upon storage.

Authors should introduce new methods in the relevant section – Methods rather than Results and Discussion. It is somewhat confusing to first read about the method used in the discussion section.

In line 288 the term “objective value” was used. To the best of my knowledge, it is not a common term, at least in the pharmaceutical literature. Does it refer to the target/dependent variable? The same stands for the term “explanatory valuable” (line 289)? Is it an explanatory/independent variable?

SOM feature maps in Fig. 5b were created “by arranging the data according to the constructed SOM shown in Fig. 3”. I am not sure if this is easily interpretable by the reader, authors should more carefully explain how the data presented in Fig. 5b was derived.

It is not clear from the presented and discussed results whether cross-validation was used for the determination of correlation coefficients for regression models.

The obtained regression models are very valuable, in terms of the potential for identification of properties of different mannitol grades that might affect the stability of prepared tablets. However, authors have failed to properly discuss the mechanical and/or physical phenomena that were identified with the regression model. Were the obtained findings expected? Proper discussion and comparison with the relevant studies are necessary to fully present the obtained results.

Author Response

We thank the reviewers for the valuable comments on our manuscript. The manuscript has been revised to address the reviewers’ concerns. The sentences that were revised are highlighted in yellow. A list of responses to the reviewers’ comments follows.

Responses to Reviewer #2’s comments

The concept is overall clearly presented; however, I have some concerns regarding the duration of storage for assessment of the stability. In my opinion, one week is a too short period of test for any general conclusion regarding the long-term tablets’ stability. On the other hand, if tablets are prone to succumb to significant changes in mechanical properties and/or disintegration time after just one week then one has to wonder what would happen after several months or years of storage.

RE: Since all components of the model tablets are stable against the humid storage condition, we thought that the tested tablet properties (TS, friability and DT) mainly depended on the moisture content of the tablets. In other ward, these values should be constant as long as the moisture content reached a level of saturation after storage under humid conditions. This issue is supported by relevant references [Chem. Pharm. Bull. 67, 284-288 (2019); Chem. Pharm. Bull. 64, 1256-1261 (2016), Drug Discov. Ther. 6, 263-268 (2012)]. Additionally, we have previously monitored hygroscopic behaviors of the model tablets occurring under the same storage condition (25oC, 75%RH) and then confirmed that moisture contents of the tablets reached a level of saturation within the first week of storage under humid conditions [Ref#10]. Taken together, we believe the change in tablet properties after 1-week storage can be used to evaluate the storage stability. The above things were fully explained in the text (Page 5, Lines 181-194).

The abstract ends with the statement that “this study could offer insight into the characteristics f mannitol”. This is too vague, authors should be more precise on what specific properties of mannitol, or properties of tablets that are made with different grades of mannitol, have been identified.

RE: We deleted the statement the reviewer pointed out and revised the text of the abstract as follows.

Page 1, Line 32-33

This study provides advanced technical knowledge to characterize the effect of different direct compaction grades of mannitol on the storage stability of tablet properties.

One general comment is that authors should omit over self-citations and provide more comparisons and/or correlations to studies of other research groups, especially in the context of properties of mannitol.

RE: According to the reviewer’s comment, we revised the Result and Discussion section and quoted relevant references concerning properties of different mannitol grade-containing tablets.

Page 10, Lines 336-343

These relationships are consistent with relevant comparative studies [6, 8]. Paul et al. [6] reported that granulated mannitol (i.e., 300DC, 400DC and 500DC) exhibit poorer compactabiity than those of spray-dried mannitol (i.e., M100, M200, 100SD and 200SD). The authors mentioned that the granulated mannitol has powder properties of larger particle size, smaller SA and lower plasticity, and then these powder properties lead to smaller bonding area of particles during compression, resulting in the poor compactability. Wagner et al. [8] also tested compactability of five different spray-dried grades of mannitol, and then reported that mannitol grades having higher SA prone to produce tablets having higher TS.

In the introduction section authors have highlighted the benefits of mannitol in the formulation of orally disintegrating tablets (ODTs) as well as its superiority over lactose. Some sentences are repetitive, especially concerning comparison to lactose, therefore this whole paragraph should be shortened. In addition, the whole first part of the introduction section is a repetition, although somewhat reformulated, of the authors’ previous publication.

RE: We took account of the comment and then revised the introduction section so as to avoid repeating the same statements.

In line 56, I suggest authors list different powder properties of directly compressible commercial grades of mannitol. Also, authors should highlight what are the critical quality attributes of ODTs in general, in order to avoid the reflection of their previous paper to a great extent.

RE: In our previous study, we investigated a wide range of powder properties of directly compressible commercial grades of mannitol [Ref#9], and then discussed fully the differences in powder properties. The previous study examined the initial level of tablet properties, while the present study investigated storage stability of tablet properties accompanying storage under humid conditions: thus, we think the research subject of the present study is distinct from the previous study. Additionally, the mannitol grades tested in this study are used for not only ODTs but also conventional pharmaceutical tablets. Thus, the target tablets of this study are not specific for ODTs. The tablet properties tested in this study (TS, friability and DT) are commonly cited as crucial quality attributes of both ODTs and conventional tablets.

Authors should include additional references, apart from their previous study, that are relevant for the presented findings. Have there been other reports on the stability testing where the samples have been conditioned for just one week? What about the selection of algorithms/models for the data analysis? From the current perspective, it seems that they were somehow randomly chosen. I suggest authors introduce more carefully the available algorithms and discuss the properties of the dataset size and data attributes that allow the usage of specific methods for data analysis. If other data analysis methods were used, even if the outcome was unsuccessful, it should be interpreted as well in order to help and provide guidance for their potential future applications for similar studies.

RE: Relative references were quoted in the Result and Discussion section as many as we can. These studies compared the initial properties of tablets made from different grades of mannitol. Please let us inform that there are no relevant reference concerning the storage test on different mannitol grade-containing tablets. Regarding ENET regression technique, we provide detailed explanation in the Method section. As the reviewer pointed out, the previous study used Lasso regresson technique, while the present study used ENET regression technique. That is because we noticed that ENET is the latest sparse modeling method compared to Lasso and ridge regressions.

Page 4, Lines 159-167

ENET is the latest sparse modeling method which is similar to the Lasso and ridge regressions [12, 13]. Sparse modeling methods feature regularized or penalized regression techniques [20]. Better regression models are constructed by shrinking the model coefficients toward zero. The validity of the estimated regression models was tested using Bayesian information criterion. These models have a great capacity to identify the crucial factors for characteristics from data sets consisting of a large number of variables. Additionally, the analysis is robust against confounding bias compared with conventional regression methods. Recently, sparse modeling methods including ENET have attracted great attention as being powerful tools for the characterization of multidimensional data. Some technical reports have applied ENET in pharmaceutical research [14, 15].

The introduction section should contain more relevant references regarding the issues on compaction properties of mannitol, stability studies, and data analysis methods, rather than already known properties of lactose and ODTs. In the current version, the last paragraph of the introduction section is just a recapitulation of what the authors have done and as such is not acceptable.

RE: We took account of the comment and revised the Introduction section. We would like to notice again that we could not find relevant reference concerning the storage test on different mannitol grade-containing tablets.

Page 2, Lines 54-63

Each grade possesses different powder properties [4, 6, 7], which substantially affect the production and quality of the tablet products in a complex manner. Therefore, a comprehensive understanding of the effects of the different commercial grades on the properties of the resulting tablet has been very important for designing tablet formulations. Several researchers have investigated different mannitol grades in terms of the characterization of powder properties and the effect on tablet properties [4, 6, 8]. For example, Paul et al. examined 11 different grades of mannitol in terms of relationships between the powder properties (e.g., particle size, surface area, plasticity) and tableting performance (e.g., tensile strength of the resulting tablets) [4, 6]. Similarly, Wagner et al. investigated the compactability of 5 different spray-dried mannitol [8].

Authors should comment on the formulation composition – why were MCC and L-HPC used as additional excipients? What about the superdisintegrant(s) that are usually used for ODTs, and that are known that can potentially impact both mechanical properties and the disintegration time of tablets?

RE: The formulation composition of the model tablets was determined by considering our previous studies. MCC and L-HPC are binder and disintegrant which are commonly used for the preparation of model tablets in our studies. We also commonly use crospovidone, superdisintegtant, especially for the preparation of ODTs. The notable characteristics of superdisintegrant is the excellent disintegration effect. However, in this case, we afraid that the disintegration effect of superdisintegrant was so strong that we could not evaluate slight difference in the disintegration properties of the test tablets. Thus, this study selected L-HPC as a disintegrant for the model tablets. Since this study aimed at not only ODTs but also conventional pharmaceutical tablets, we believe using L-HPC is not significant matter for completing this research work.

Is there any specific reason for manufacturing at such a large scale – three hundred kilograms per batch?

RE: We apologize that we made a mistake in the text of sample preparation. Correctly, three hundred “grams” of the powder blend were prepared for each model tablets. We have already corrected this issue. Thank you for your pointing out my mistake.

There are some technical issues with Eq. 1.

How were the parameters for the friability test selected?

RE: Eq1 is used for calculation of tensile strength. The tensile strength was calculated from the values of hardness, diameter and thickness of the tablets. We revised the text as follows.

Page 3, Line 119-120.

…where F is the hardness (the maximum diametric crushing force) and d and t are the diameter and thickness of the tablet, respectively..

Authors have stated, in line 137, that there was triplicate data acquisition for 15 different grades of mannitol. Triplicate would mean that the same conditions were used for each batch for three times. However, each batch was compressed at different compression pressures to produce three different tablets’ hardness values. I suggest authors to define more precisely what was varied and what was repeated.

RE: This study prepared one batch for each mannitol grade-containing tablet. The measurement of tablet properties was repeated three times. To avoid confusion for readers, the following sentence written in the original text was removed from the revised text. “The number in the training data set was 45 (15 different grades of mannitol and triplicate data acquisition).”

The authors should state the number of tablets per batch that were used to test tensile strength, friability, and disintegration time. In line 186 it is stated that measurements were made in triplicate which is insufficient, especially in the case of disintegration time and friability.

RE: Regarding the measurement of tensile strength and disintegration time, a single measurement needs only one tablet. Since the measurement was repeated three times, a total of three tablets are used for each tablet property. As for the measurement of friability, approximately 4 g of tablets were used in a single measurement. Since the measurement of friability was also repeated three times, so a total of approximately 12 g of tablets were used.

Also, we have revised the text as follows.

Page 6, Lines 232-233

The xpre corresponds to the mean value of each grade of mannitol tablet for the variable measurement, which was repeated three times.   

A comparison between the results of mannitol tablets characterization with the relevant references (other from authors’ own) should have been made.

What is the expected reason for the reduction in tablets’ tensile strength upon one week of storage? Are there any other reports on the same matter? Why did tablets made of 400DC have the highest change in tablets tensile strength and friability upon storage?

A proper discussion of the obtained results, in terms of the effect of storage on tablets properties, should contain proposed mechanisms and/or explanations of the obtained findings. In the current version of the manuscript, authors are just describing in more detail the results that are presented in Figure 1.

RE: To the best of our knowledge, this is the first technical report on the effects of different mannitol grade on storage stability of tablet properties. Thus, the main purpose of this study is to characterize the effect of different mannitol grades on storage stability of the tablet properties and to screen the crucial powder properties of mannitol for the change in tablet properties accompanying storage. We realize that, at present, many things remain unclear and there are plenty room for investigation, because this study is in the early stage. We admit that we don’t clarify as yet the reason why 400DC have the highest change in tablet’s tensile strength and friability upon storage. In the next study, we going to investigate this issue more in detail. So, we would like to deal with the suggestion as the research subject of our next study.

I disagree with the authors that the introduced measure in Eq.2 removes the influence of the tablet hardness on the obtained data. It just represents the % of retention of a tablet property upon storage.

RE: Since tablet properties, especially TS and DT, were changed proportionally according to the tablet hardness, we employed the % retention of the tablet properties as an indicator to evaluate the storage stability of each tablet. We revised the text as follows.

Page 6, Lines 221-227

Subsequently, we characterize the changes in the tablet properties of each model tablet after storage. We calculated the % retention of the tablet properties from the prestorage level as an indicator to evaluate the storage stability of each tablet. As shown in Fig.1, tablet properties, especially TS and DT, were changed proportionally according to the tablet hardness. Since the % retention is a parameter that removes the influence of absolute values of tablet properties from the raw data, we assumed that the all data can be evaluated regardless of the tablet hardness by using this parameter.

Authors should introduce new methods in the relevant section – Methods rather than Results and Discussion. It is somewhat confusing to first read about the method used in the discussion section.

RE: We moved the explanation about SOM and ENET to Method section from Results and Discussion section.

Page 4, Lines 144-152

The Kohonen SOM is a feedforward-type neural network model [11]. This analysis allows multidimensional data to be turned into a two-dimensional map. This is considered to be a powerful tool for characterizing multidimensional data, especially clustering data. The typical structure of a SOM is composed of one input layer and one output layer. Nodes containing parametric reference vectors are arranged in a regular pattern in the output layer. The SOM is constructed via an unsupervised competitive learning approach. As a result of competitive learning, the array of the output nodes ultimately reflects the patterns of learning data sets. The adjacent nodes possess closer reference vectors to each other; thus, the distance between the nodes can be regarded as the degree of similarity between them

Page 4, Lines 159-167

The validity of the estimated regression models was tested using Bayesian information criterion. ENET is the latest sparse modeling method which is similar to the Lasso and ridge regressions [12, 13]. Sparse modeling methods feature regularized or penalized regression techniques [20]. Better regression models are constructed by shrinking the model coefficients toward zero. These models have a great capacity to identify the crucial factors for characteristics from data sets consisting of a large number of variables. Additionally, the analysis is robust against confounding bias compared with conventional regression methods. Recently, sparse modeling methods including ENET have attracted great attention as being powerful tools for the characterization of multidimensional data. Some technical reports have applied ENET in pharmaceutical research [14, 15].

In line 288 the term “objective value” was used. To the best of my knowledge, it is not a common term, at least in the pharmaceutical literature. Does it refer to the target/dependent variable? The same stands for the term “explanatory valuable” (line 289)? Is it an explanatory/independent variable?

RE: We deleted the terms of “objective value” and “explanatory valuable” and then revised the text as follows.

Page 10, Line 345-347

The analysis was performed using ENET regression modeling. According to the analysis, we tried to identify the crucial powder properties for the % retention from the prestorage level in the tablet properties.

SOM feature maps in Fig. 5b were created “by arranging the data according to the constructed SOM shown in Fig. 3”. I am not sure if this is easily interpretable by the reader, authors should more carefully explain how the data presented in Fig. 5b was derived.

RE: We revised the explanation about how to create SOM feature map shown in Fig.5B.

Page10, Lines 328-332

In addition, the SOM feature maps of the powder and tablet properties were presented in Fig.5B. Since the input data for the SOM clustering contained not only training variables (% retention values), but also powder and tablet properties, their SOM feature maps could be created from the SOM constructed before.

It is not clear from the presented and discussed results whether cross-validation was used for the determination of correlation coefficients for regression models.

The obtained regression models are very valuable, in terms of the potential for identification of properties of different mannitol grades that might affect the stability of prepared tablets. However, authors have failed to properly discuss the mechanical and/or physical phenomena that were identified with the regression model. Were the obtained findings expected? Proper discussion and comparison with the relevant studies are necessary to fully present the obtained results.

RE: In this study, the validity of ENET regression models were tested by Bayesian information criterion not by cross validation. So, we did not perform cross validation in making the scatterplots.  

Also, this is the first technical report concerning storage stability of different grades of mannitol. To date, there are no relevant studies conducted by the other researchers from this perspective. Thus, before starting this study, we did not have any prospect about crucial powder properties of mannitol grades for storage stability of properties of the resulting tablets. We realize that, at present, there are plenty room for investigation about the relationships between powder properties and storage stability of the resulting tablets. In near future, we are going to investigate the significant relationships observed in this study one by one (e.g., relationship between D50 and storage stability of TS). Thus, we would like to deal with this issue as the research subject of our next study.

Round 2

Reviewer 1 Report

The authors have carefully considered all of the feedback provided by this reviewer and have developed the revised version of the manuscript with the feedback duly considered with the exception of one aspect:

Page 3 line 103- 106: authors must indicate solid fraction produced by compressing the different mannitol blends.  This data (to be put into the results and discussion) may help further explain how differences in mannitol particle properties result in tablets with different initial and storage properties when compressed to specific hardness. (From previous review report).

The authors had responded to this comment around mannitol particle characterisation by microscopy. Tablet porosity or solid fraction may be a contributor to the differences seen and may be associated with the range of particle size distributions seen for the various  grades of mannitol studied.  So although the authors did not reconsider their tablet manufacturing with different mannitol grades in estimating how the interplay between  mannitol particle size and applied force/resultant  tablet crushing strength reflect tablet porosity/solid fraction, they have noted that there is an influence of D50 and SSA, which could modulate solid fraction/tablet porosity in tablets manufactured to target hardness by adjusting applied force (lines 369-382).  So, as the authors have indirectly recognised  this aspect, I will not push the point further (though I do wish they had  pursued the determination of tablet solid fraction!  It might have further indicated the mechanism by which the D50 and SSA influence time-dependent properties in the tablets.).

Overall, the manuscript is much improved and the work of the authors is worthy of further consideration for publication.

Reviewer 2 Report

The authors have addressed all comments/suggestions in a satisfactory manner. 

There is a spelling error in the supplementary material: Parteck is spelled incorrectly.